# Erdős Goes Neural: an Unsupervised Learning Framework for Combinatorial Optimization on Graphs

**Nikolaos Karalias**
EPFL
nikolaos.karalias@epfl.ch

**Andreas Loukas**
EPFL
andreas.loukas@epfl.ch

## Abstract

Combinatorial optimization (CO) problems are notoriously challenging for neural networks, especially in the absence of labeled instances. This work proposes an unsupervised learning framework for CO problems on graphs that can provide integral solutions of certified quality. Inspired by Erdős' probabilistic method, we use a neural network to parametrize a probability distribution over sets. Crucially, we show that when the network is optimized w.r.t. a suitably chosen loss, the learned distribution contains, with controlled probability, a low-cost integral solution that obeys the constraints of the combinatorial problem. The probabilistic proof of existence is then derandomized to decode the desired solutions. We demonstrate the efficacy of this approach to obtain valid solutions to the maximum clique problem and to perform local graph clustering. Our method achieves competitive results on both real datasets and synthetic hard instances.

## 1 Introduction

Combinatorial optimization (CO) includes a wide range of computationally hard problems that are omnipresent in scientific and engineering fields. Among the viable strategies to solve such problems are neural networks, which were proposed as a potential solution by Hopfield and Tank [30]. Neural approaches aspire to circumvent the worst-case complexity of NP-hard problems by only focusing on instances that appear in the data distribution.

Since Hopfield and Tank, the advent of deep learning has brought new powerful learning models, reviving interest in neural approaches for combinatorial optimization. A prominent example is that of graph neural networks (GNNs) [28, 60], whose success has motivated researchers to work on CO problems that involve graphs [35, 87, 39, 27, 43, 53, 7, 56] or that can otherwise benefit from utilizing a graph structure in the problem formulation [69] or the solution strategy [27]. The expressive power of graph neural networks has been the subject of extensive research [82, 47, 17, 59, 58, 8, 26]. Encouragingly, GNNs can be Turing universal in the limit [46], which motivates their use as general-purpose solvers.

Yet, despite recent progress, CO problems still pose a significant challenge to neural networks. Successful models often rely on supervision, either in the form of labeled instances [45, 62, 35] or of expert demonstrations [27]. This success comes with drawbacks: obtaining labels for hard problem instances can be computationally infeasible [86], and direct supervision can lead to poor generalization [36]. Reinforcement learning (RL) approaches have also been used for both classical CO problems [16, 87, 85, 41, 20, 38, 7] as well as for games with large discrete action spaces, like Starcraft [75] and Go [64]. However, not being fully-differentiable, they tend to be harder and more time consuming to train.

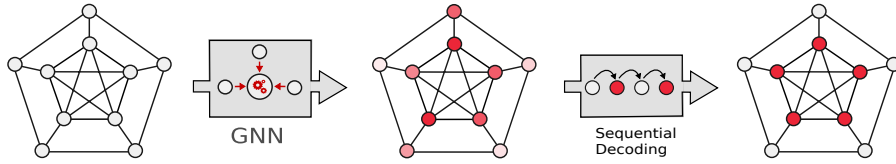

Figure 1: Illustration of the "Erdős goes neural" pipeline. First, a differentiable loss is derived for a given problem using the probabilistic method. Next, a GNN is trained in an unsupervised way using the derived loss to output a probability distribution over the nodes, essentially providing a probabilistic certificate for the existence of a low cost feasible solution. At inference time, a discrete solution satisfying the certificate is obtained in a sequential and deterministic manner by the method of conditional expectation.

An alternative to these strategies is unsupervised learning, where the goal is to model the problem with a differentiable loss function whose minima represent the discrete solution to the combinatorial problem [65, 10, 2, 3, 69, 85]. Unsupervised learning is expected to aid in generalization, as it allows the use of large unlabeled datasets, and it is often envisioned to be the long term goal of artificial intelligence. However, in the absence of labels, deep learning faces practical and conceptual obstacles. Continuous relaxations of objective functions from discrete problems are often faced with degenerate solutions or may simply be harder to optimize. Thus, successful training hinges on empirically-identified correction terms and auxiliary losses [10, 3, 71]. Furthermore, it is especially challenging to decode valid (with respect to constraints) discrete solutions from the soft assignments of a neural network [45, 69], especially in the absence of complete labeled solutions [62].

Our framework aims to overcome some of the aforementioned obstacles of unsupervised learning: *it provides a principled way to construct a differentiable loss function whose minima are guaranteed to be low-cost valid solutions of the problem*. Our approach is inspired by Erdős' probabilistic method and entails two steps: First, we train a GNN to produce a distribution over subsets of nodes of an input graph by minimizing a probabilistic penalty loss function. Successfully optimizing our loss is guaranteed to yield good integral solutions that obey the problem constraints. After the network has been trained, we employ a well-known technique from randomized algorithms to sequentially and deterministically decode a valid solution from the learned distribution. The procedure is schematically illustrated in Figure 1.

We demonstrate the utility of our method in two NP-hard graph-theoretic problems: the *maximum clique* problem [12] and a *constrained min-cut* problem [15, 66] that can perform local graph clustering [4, 77]. In both cases, our method achieves competitive results against neural baselines, discrete algorithms, and mathematical programming solvers. Our method outperforms the CBC solver (provided with Google's OR-Tools), while also remaining competitive with the SotA commercial solver Gurobi 9.0 [29] on larger instances. Finally, our method outperforms both neural baselines and well-known local graph clustering algorithms in its ability to find sets of good conductance, while maintaining computational efficiency. [1]

## 2   Related work and background

### 2.1   Neural networks for combinatorial optimization

Most neural approaches to CO are supervised. One of the first modern neural networks were the Pointer Networks [74], which utilized a sequence-to-sequence model for the travelling salesman problem (TSP). Since then, numerous works have combined GNNs with various heuristics and search procedures to solve classical CO problems, such as quadratic assignment [53], graph matching [6], graph coloring [43], TSP [45, 35], and even sudoku puzzles [54]. Another fruitful direction has been the fusion with solvers. For example, Neurocore [61] incorporates an MLP to a SAT solver to enhance variable branching decisions, whereas Gasse et al. [27] learn branching approximations by a GNN

and imitation learning. Further, Wang et al. [78] include an approximate SDP satisfiability solver as a neural network layer and Vlastelica et al. [76] incorporate exact solvers within a differentiable architecture by smoothly interpolating the solver's piece-wise constant output. Unfortunately, the success of supervised approaches hinges on building large training sets with already solved hard instances, resulting in a chicken and egg situation. Moreover, since it is hard to efficiently sample unbiased and representative labeled instances of an NP-hard problem [86], labeled instance generation is likely not a viable long-term strategy either.

Training neural networks without labels is generally considered to be more challenging. One possibility is to use RL: Khalil et al. [38] combine Q-Learning with a greedy algorithm and structure2vec embeddings to solve max-cut, minimum vertex cover, and TSP. Q-Learning is also used in [7] for the maximum common subgraph problem. On the subject of TSP, the problem was also solved with policy gradient learning combined with attention [41, 20, 9]. Attention is ubiquitous in problems that deal with sequential data, which is why it has been widely used with RL for the problem of vehicle routing [25, 51, 55, 33]. Another interesting application of RL is the work of Yolcu and Poczos [87], where the REINFORCE algorithm is employed in order to learn local search heuristics for the SAT problem. This is combined with curriculum learning to improve stability during training. Finally, Chen and Tian [16] use actor-critic learning to iteratively improve complete solutions to combinatorial problems. Though a promising research direction, deep RL methods are far from ideal, as they can be sample inefficient and notoriously unstable to train—possibly due to poor gradient estimates, dependence on initial conditions, correlations present in the sequence of observations, bad rewards, sub-optimal hyperparameters, or poor exploration [68, 52, 31, 48].

The works that are more similar to ours are those that aim to train neural networks in a differentiable and end-to-end manner: Toenshoff et al. [69] model CO problems in terms of a constraint language and utilize a recurrent GNN, where all variables that coexist in a constraint can exchange messages. Their model is completely unsupervised and is suitable for problems that can be modeled as maximum constraint satisfaction problems. For other types of problems, like independent set, the model relies on empirically selected loss functions to solve the task. Amizadeh et al. [2, 3] train a GNN in an unsupervised manner to solve the circuit-SAT and SAT problems by minimizing an appropriate energy function. Finally, Yao et al. [85] train a GNN for the max-cut problem on regular graphs without supervision by optimizing a smooth relaxation of the cut objective and policy gradient.

Our approach innovates from previous works in the following ways: it enables training a neural network in an unsupervised, differentiable, and end-to-end manner, while also ensuring that identified solutions will be integral and will satisfy problem constraints. Crucially, this is achieved in a simple and mathematically-principled way, without resorting to continuous relaxations, regularization, or heuristic corrections of improper solutions. In addition, our approach does not necessitate polynomial-time reductions, but solves each problem directly.

## 2.2 Background: the probabilistic method

The probabilistic method is a nonconstructive proof method pioneered by Paul Erdős. It is used to demonstrate the existence of objects with desired combinatorial properties [1], [22], [67] but has also served as the foundation for important algorithms in the fields of computer science and combinatorial optimization [49] [57].

Let us consider the common didactic example of the maximum cut problem on a simple undirected graph. The goal is to bipartition the nodes of the graph in such a way that the number of edges with endpoints in both partitions (i.e., the cardinality of the cut-set) is maximized. For simplicity we will refer to the cardinality of the cut-set as the cut. Suppose we decide the bipartition based on a fair coin flip, i.e., we split the nodes of the graph by assigning them to a heads or a tails set. An edge belongs to the cut-set when its endpoints belong to different sets. This happens with probability $1/2$, which implies that *the expected cut* will be equal to half of the edges of the graph. Thus, by Markov's inequality and given that the cut is non-negative, it follows that there exists a bipartitioning that contains *at least half* of the edges of the graph.

To obtain such a solution deterministically, we will utilize the method of conditional expectation [57]: we sequentially visit every node $v_i$ in the graph and we compute the expected cut conditioned on $v_i$ belonging to the heads or tails set (together with all the decisions made until the $i$-th step) and add $v_i$ to the set (heads or tails) that yields smaller conditional expected cut. Since the (conditional)

expectation can only improve at every step, the sets recovered are guaranteed to cut at least half the edges of the graph, as proved earlier.

Our goal is to re-purpose this classic approach to tackle combinatorial optimization problems with deep learning. In this work, instead of using a naive probability assignment like in the maxcut example, the probability distribution is learned by a GNN which allows us to obtain higher quality solutions. Additionally, we show how this argument may be extended to incorporate constraints within the learning paradigm.

# 3 The Erdős probabilistic method for deep learning

We focus on combinatorial problems on weighted graphs $G = (V, E, w)$ that are modelled as constrained optimization problems admitting solutions that are node sets:

$$\min_{S \subseteq V} \; f(S; G) \quad \text{subject to} \quad S \in \Omega. \tag{1}$$

Above, $\Omega$ is a family of sets having a desired property, such as forming a clique or covering all nodes. This yields a quite general formulation that can encompass numerous classical graph-theoretic problems, such as the maximum clique and minimum vertex cover problems.

## 3.1 The "Erdős Goes Neural" pipeline

Rather than attempting to optimize the non-differentiable problem (1) directly, we propose to train a GNN to identify distributions of solutions with provably advantageous properties. Our approach is inspired by Erdős' probabilistic method, a well known technique in the field of combinatorics that is used to prove the existence of an object with a desired combinatorial property.

As visualized in Figure 1, our method consists of three steps:

1. Construct a GNN $g_\theta$ that outputs a distribution $\mathcal{D} = g_\theta(G)$ over sets.

2. Train $g_\theta$ to optimize the probability that there exists a valid $S^* \sim \mathcal{D}$ of small cost $f(S^*; G)$.

3. Deterministically recover $S^*$ from $\mathcal{D}$ by the method of conditional expectation.

There are several possibilities in instantiating $\mathcal{D}$. We opt for the simplest and suppose that the decision of whether $v_i \in S$ is determined by a Bernoulli random variable $x_i$ of probability $p_i$. The network can trivially parametrize $\mathcal{D}$ by computing $p_i$ for every node $v_i$. Keeping the distribution simple will aid us later on to tractably control relevant probability estimates.

Next, we discuss how $g_\theta$ can be trained (Section 3.2) and how to recover $S^*$ from $\mathcal{D}$ (Section 3.3).

## 3.2 Deriving a probabilistic loss function

The main challenge of our method lies in determining how to tractably and differentiably train $g_\theta$. Recall that our goal is to identify a distribution that contains low-cost and valid solutions.

### 3.2.1 The probabilistic loss

Aiming to build intuition, let us first consider the unconstrained case. To train the network, we construct a loss function $\ell(\mathcal{D}; G)$ that abides to the following property:

$$P(f(S; G) < \ell(\mathcal{D}; G)) > t \quad \text{with} \quad \mathcal{D} = g_\theta(G). \tag{2}$$

Any number of tail inequalities can be used to instantiate such a loss, depending on the structure of $f$. If we only assume that $f$ is non-negative, Markov's inequality yields

$$\ell(\mathcal{D}; G) \triangleq \frac{\mathbb{E}\left[f(S; G)\right]}{1 - t} \quad \text{for any} \quad t \in [0, 1).$$

If the expectation cannot be computed in closed-form, then any upper bound also suffices.

The main benefit of approaching the problem in this manner is that the surrogate (and possibly differentiable) loss function $\ell(\mathcal{D}; G)$ can act as a certificate for the existence of a good set in the support of $\mathcal{D}$. To illustrate this, suppose that one has trained $g_\theta$ until the loss is sufficiently small, say $\ell(\mathcal{D}; G) = \epsilon$. Then, by the probabilistic method, there exists with strictly positive probability a set $S^*$ in the support of $\mathcal{D}$ whose cost $f(S^*; G)$ is at most $\epsilon$.

### 3.2.2 The probabilistic penalty loss

To incorporate constraints, we take inspiration from penalty methods in constrained optimization and add a term to the loss function that penalizes deviations from the constraint.

Specifically, we define the probabilistic penalty function $f_p(S; G) \triangleq f(S; G) + \mathbf{1}_{S \notin \Omega}\beta$, where $\beta$ is a scalar. The expectation of $f_p$ yields the probabilistic penalty loss:

$$\ell(\mathcal{D}, G) \triangleq \mathbb{E}\left[f(S; G)\right] + P(S \notin \Omega)\beta. \tag{3}$$

We prove the following:

**Theorem 1.** Fix any $\beta > \max_S f(S; G)$ and let $\ell(\mathcal{D}, G) = \epsilon < \beta$. With probability at least $t$, set $S^* \sim \mathcal{D}$ satisfies

$$f(S^*; G) < \ell(\mathcal{D}; G)/(1 - t) \ \ \text{and} \ \ S^* \in \Omega,$$

under the condition that $f$ is non-negative.

Hence, similar to the unconstrained case, the penalized loss acts as a certificate for the existence of a low-cost set, but now the set is also guaranteed to abide to the constrains $\Omega$. The main requirement for incorporating constraints is to be able to differentiably compute an upper estimate of the probability $P(S \notin \Omega)$. A worked out example of how $P(S \notin \Omega)$ can be controlled is provided in Section 4.1.

### 3.2.3 The special case of linear box constraints

An alternative construction can be utilized when problem (1) takes the following form:

$$\min_{S \subseteq V} \ f(S; G) \quad \text{subject to} \quad \sum_{v_i \in S} a_i \in [b_l, b_h], \tag{4}$$

with $a_i$, $b_l$, and $b_h$ being non-negative scalars.

We tackle such instances with a two-step approach. Denote by $\mathcal{D}^0$ the distribution of sets predicted by the neural network and let $p_1^0, \ldots, p_n^0$ be the probabilities that parametrize it. We rescale these probabilities such that the constraint is satisfied in expectation:

$$\sum_{v_i \in V} a_i p_i = \frac{b_l + b_h}{2}, \quad \text{where} \quad p_i = \text{clamp}\left(c\, p_i^0, 0, 1\right) \quad \text{and} \quad c \in \mathbb{R}.$$

Though non-linear, the aforementioned feasible re-scaling can be carried out by a simple iterative scheme (detailed in Section D.2). If we then proceed as in Section 3.2.1 by utilizing a probabilistic loss function that guarantees the existence of a good unconstrained solution, we have:

**Theorem 2.** Let $\mathcal{D}$ be the distribution obtained after successful re-scaling of the probabilities. For any (unconstrained) probabilistic loss function that abides to $P(f(S; G) < \ell(\mathcal{D}; G)) > t$, set $S^* \sim \mathcal{D}$ satisfies $f(S^*; G) < \ell(\mathcal{D}; G)$ and $\sum_{v_i \in S^*} a_i \in [b_l, b_h]$, with probability at least $t - 2\exp\left(-(b_h - b_l)^2 / \sum_i 2a_i^2\right)$.

Section 4.2 presents a worked-out example of how Theorem 2 can be applied.

## 3.3 Retrieving integral solutions

A simple way to retrieve a low cost integral solution $S^*$ from the learned distribution $\mathcal{D}$ is by monte-carlo sampling. Then, if $S^* \sim \mathcal{D}$ with probability $t$, the set can be found within the first $k$ samples with probability at least $1 - (1 - t)^k$. However, our goal is to deterministically obtain $S^*$ so we will utilize the method of conditional expectation that was introduced in Section 2.2.

Let us first consider the unconstrained case. Given $\mathcal{D}$, the goal is to identify a set $S^*$ that satisfies $f(S^*; G) \leq \mathbb{E}\left[f(S; G)\right]$. To achieve this, one starts by sorting $v_1, \ldots, v_n$ in order of decreasing probabilities $p_i$. Let $S_{\text{reject}} = \varnothing$ be the set of nodes not accepted in the solution. Set $S^* = \varnothing$ is then iteratively updated one node at a time, with $v_i$ being included to $S^*$ in the $i$-th step if $\mathbb{E}\left[f(S; G) \mid S^* \subset S, \ S \cap S_{\text{reject}} = \varnothing, \ \text{and} \ v_i \in S\right] < \mathbb{E}\left[f(S; G) \mid S^* \subset S, \ S \cap S_{\text{reject}} = \varnothing, \ \text{and} \ v_i \notin S\right]$. This sequential decoding works because the conditional expectation never increases.

In the case of the probabilistic penalty loss, the same procedure is applied w.r.t. the expectation of $f_p(S; G)$. The latter ensures that the decoded set will match the claims of Theorem 1. For the method of Section 3.2.3, a sequential decoding can guarantee either that the cost of $f(S^*; G)$ is small or that the constraint is satisfied.

# 4   Case studies

This section demonstrates how our method can be applied to two well known NP-hard problems: the *maximum clique* [12] and the *constrained minimum cut* [15] problems.

## 4.1   The maximum clique problem

A clique is a set of nodes such that every two distinct nodes are adjacent. The maximum clique problem entails identifying the clique of a given graph with the largest possible number of nodes:

$$\min_{S \subseteq V} -w(S) \quad \text{subject to} \quad S \in \Omega_{\text{clique}}, \tag{5}$$

with $\Omega_{clique}$ being the family of cliques of graph $G$ and $w(S) = \sum_{v_i, v_j \in S} w_{ij}$ being the weight of $S$. Optimizing $w(S)$ is a generalization of the standard cardinality formulation to weighted graphs. For simple graphs, both weight and cardinality formulations yield the same minimum.

We can directly apply the ideas of Section 3.2.2 to derive a probabilistic penalty loss:

**Corollary 1.** Fix positive constants $\gamma$ and $\beta$ satisfying $\max_S w(S) \leq \gamma \leq \beta$ and let $w_{ij} \leq 1$. If

$$\ell_{\text{clique}}(\mathcal{D}, G) \triangleq \gamma - (\beta + 1) \sum_{(v_i, v_j) \in E} w_{ij} p_i p_j + \frac{\beta}{2} \sum_{v_i \neq v_j} p_i p_j \leq \epsilon$$

then, with probability at least $t$, set $S^* \sim \mathcal{D}$ is a clique of weight $w(S^*) > \gamma - \epsilon/(1-t)$.

The loss function $\ell_{\text{clique}}$ can be evaluated in linear time w.r.t. the number of edges of $G$ by rewriting the rightmost term as $\sum_{v_i \neq v_j} p_i p_j = (\sum_{v_i \in V} p_i)^2 - \sum_{(v_i, v_j) \in E} 2p_i p_j$.

A remark.   One may be tempted to fix $\beta \to \infty$, such that the loss does not feature any hyper-parameters. However, with mini-batch gradient descent it can be beneficial to tune the contribution of the two terms in the loss to improve the optimization. This was also confirmed in our experiments, where we selected the relative weighting according to a validation set.

**Decoding cliques.**   After the network is trained, valid solutions can be decoded sequentially based on the procedure of Section 3.3. The computation can also be sped up by replacing conditional expectation evaluations (one for each node) by a suitable upper bound. Since the clique property is maintained at every point, we can also efficiently decode cliques by sweeping nodes (in the order of larger to smaller probability) and only adding them to the set when the clique constraint is satisfied.

## 4.2   Graph partitioning

The simplest partitioning problem is the minimum cut: find set $S \subset V$ such that $\text{cut}(S) = \sum_{v_i \in S, \, v_j \notin S} w_{ij}$ is minimized. Harder variants of partitioning aim to provide control on partition balance, as well as cut weight. We consider the following constrained min-cut problem:

$$\min_S \text{cut}(S) \quad \text{subject to} \quad \text{vol}(S) \in [v_l, v_h],$$

where the volume $\text{vol}(S) = \sum_{v_i \in S} d_i$ of a set is the sum of the degrees of its nodes.

The above can be shown to be NP-hard [32] and exhibits strong connections with other classical formulations: it is a volume-balanced graph partitioning problem [5] and can be used to minimize graph conductance [18] by scanning through solutions in different volume intervals and selecting the one whose cut-over-volume ratio is the smallest (this is how we test it in Section 5).

We employ the method described in Section 3.2.3 to derive a probabilistic loss function:

**Corollary 2.** Let the probabilities $p_1, \ldots, p_n$ giving rise to $\mathcal{D}$ be re-scaled such that $\sum_{v_i \in V} d_i p_i = \frac{v_l + v_h}{2}$ and, further, fix $\ell_{\text{cut}}(\mathcal{D}; G) \triangleq \sum_{v_i \in V} d_i p_i - 2 \sum_{(v_i, v_j) \in E} p_i p_j w_{ij}$. Set $S^* \sim \mathcal{D}$ satisfies

$$\text{cut}(S^*) < \ell_{\text{cut}}(\mathcal{D}; G)/(1-t) \quad \text{and} \quad \text{vol}(S^*) \in [v_l, v_h],$$

with probability at least $t - 2 \exp\left(-(v_h - v_l)^2 / \sum_i 2d_i^2\right)$.

The derived loss function $\ell_{\text{cut}}$ can be computed efficiently on a sparse graph, as its computational complexity is linear on the number of edges.

**Decoding clusters.** Retrieving a set that respects Corollary 2 can be done by sampling. Alternatively, the method described in Section 3.3 can guarantee that the identified cut is at most as small as the one certified by the probabilistic loss. In the latter case, the linear box constraint can be practically enforced by terminating before the volume constraint gets violated.

## 5 Empirical evaluation

We evaluate our approach in its ability to find large cliques and partitions of good conductance.

### 5.1 Methods

We refer to our network as Erdős' GNN, paying tribute to the pioneer of the probabilistic method that it is inspired from. Its architecture comprises of multiple layers of the Graph Isomorphism Network (GIN) [81] and a Graph Attention (GAT) [73] layer. Furthermore, each convolution layer was equipped with skip connections, batch normalization and graph size normalization [21]. In addition to a graph, we gave our network access to a one-hot encoding of a randomly selected node, which encourages locality of solutions, allows for a trade-off between performance and efficiency (by rerunning the network with different samples), and helps the network break symmetries [63]. Our network was trained with mini-batch gradient descent, using the Adam optimizer [40] and was implemented on top of the pytorch geometric API [23].

*Maximum clique.* We compared against three neural networks, three discrete algorithms, and two integer-programming solvers: The neural approaches comprised of *RUN-CSP*, *Bomze GNN*, and *MS GNN*. The former is a SotA unsupervised network incorporating a reduction to independent set and a post-processing of invalid solutions with a greedy heuristic. The latter two, though identical in construction to Erdős' GNN, were trained based on standard smooth relaxations of the maximum clique problem with a flat 0.5-threshold discretization [50, 11]. Since all these methods can produce multiple outputs for the same graph (by rerunning them with different random node attributes), we fix two time budgets for RUN-CSP and Erdős' GNN, that we refer to as "fast" and "accurate" and rerun them until the budget is met (excluding reduction costs). On the other hand, the Bomze and MS GNNs are rerun 25 times, since further repetitions did not yield relevant improvements. We considered the following algorithms: the standard *Greedy MIS Heur.* which greedily constructs a maximal independent set on the complement graph, *NX MIS approx.* [13], and *Toenshoff-Greedy* [69]. Finally, we formulated the maximum clique in integer form [12] and solved it with *CBC* [34] and *Gurobi* 9.0 [29], an open-source solver provided with Google's OR-Tools package and a SotA commercial solver. We should stress that our evaluation does not intend to establish SotA results (which would require a more exhaustive comparison), but aims to comparatively study the weaknesses and strengths of key unsupervised approaches.

*Local partitioning.* We compared against two neural networks and four discrete algorithms. To the extent of our knowledge, no neural approach for constrained partitioning exists in the literature. Akin to maximum clique, we built the *L1 GNN* and *L2 GNN* to be identical to Erdős' GNN and trained them based on standard smooth $\ell_1$ and $\ell_2$ relaxations of the cut combined with a volume penalty. On the other hand, a number of algorithms are known for finding small-volume sets of good conductance. We compare to well-known and advanced algorithms [24]: *Pagerank-Nibble* [4], Capacity Releasing Diffusion (*CRD*) [77], Max-flow Quotient-cut Improvement (*MQI*) [42] and *Simple-Local* [72].

### 5.2 Data

Experiments for the maximum clique were conducted in the IMDB, COLLAB [37, 84] and TWITTER [44] datasets, listed in terms of increasing graph size. Further experiments were done on graphs

| | IMDB | COLLAB | TWITTER |
|---|---|---|---|
| Erdős' GNN (fast) | 1.000 (0.08 s/g) | 0.982 ± 0.063 (0.10 s/g) | **0.924 ± 0.133 (0.17 s/g)** |
| Erdős' GNN (accurate) | 1.000 (0.10 s/g) | 0.990 ± 0.042 (0.15 s/g) | 0.942 ± 0.111 (0.42 s/g) |
| RUN-CSP (fast) | 0.823 ± 0.191 (0.11 s/g) | 0.912 ± 0.188 (0.14 s/g) | 0.909 ± 0.145 (0.21 s/g) |
| RUN-CSP (accurate) | 0.957 ± 0.089 (0.12 s/g) | 0.987 ± 0.074 (0.19 s/g) | 0.987 ± 0.063 (0.39 s/g) |
| Bomze GNN | 0.996 ± 0.016 (0.02 s/g) | *0.984 ± 0.053 (0.03 s/g)* | *0.785 ± 0.163 (0.07 s/g)* |
| MS GNN | 0.995 ± 0.068 (0.03 s/g) | *0.938 ± 0.171 (0.03 s/g)* | *0.805 ± 0.108 (0.07 s/g)* |
| NX MIS approx. | 0.950 ± 0.071 (0.01 s/g) | 0.946 ± 0.078 (1.22 s/g) | 0.849 ± 0.097 (0.44 s/g) |
| Greedy MIS Heur. | 0.878 ± 0.174 (1e-3 s/g) | 0.771 ± 0.291 (0.04 s/g) | 0.500 ± 0.258 (0.05 s/g) |
| Toenshoff-Greedy | 0.987 ± 0.050 (1e-3 s/g) | 0.969 ± 0.087 (0.06 s/g) | **0.917 ± 0.126 (0.08 s/g)** |
| CBC (1s) | 0.985 ± 0.121 (0.03 s/g) | 0.658 ± 0.474 (0.49 s/g) | 0.107 ± 0.309 (1.48 s/g) |
| CBC (5s) | 1.000 (0.03 s/g) | 0.841 ± 0.365 (1.11 s/g) | 0.198 ± 0.399 (4.77 s/g) |
| Gurobi 9.0 (0.1s) | **1.000 (1e-3 s/g)** | 0.982 ± 0.101 (0.05 s/g) | 0.803 ± 0.258 (0.21 s/g) |
| Gurobi 9.0 (0.5s) | 1.000 (1e-3 s/g) | 0.997 ± 0.035 (0.06 s/g) | 0.996 ± 0.019 (0.34 s/g) |
| Gurobi 9.0 (1s) | 1.000 (1e-3 s/g) | 0.999 ± 0.015 (0.06 s/g) | **1.000 (0.34 s/g)** |
| Gurobi 9.0 (5s) | 1.000 (1e-3 s/g) | **1.000 (0.06 s/g)** | 1.000 (0.35 s/g) |

Table 1: Test set approximation ratios for all methods on real-world datasets. For solvers, time budgets are listed next to the name. Pareto-optimal solutions are indicated in bold, whereas italics indicate constraint violation (we report the results only for correctly solved instances).

generated from the RB model [80], that has been specifically designed to generate challenging problem instances. We worked with three RB datasets: a training set containing graphs of up to 500 nodes [69], a newly generated test set containing graphs of similar size, and a set of instances that are up to 3 times larger [79, 45, 69]. On the other hand, to evaluate partitioning, we focused on the FACEBOOK [70], TWITTER, and SF-295 [83] datasets, with the first being a known difficult benchmark. More details can be found in the Appendix.

*Evaluation.* We used a 60-20-20 split between training, validation, and test for all datasets, except for the RB model data (details in paragraph above). Our baselines often require the reduction of maximum clique to independent set, which we have done when necessary. The reported time costs factor in the cost of reduction. During evaluation, for each graph, we sampled multiple inputs, obtained their solutions, and kept the best one. This was repeated for all neural approaches and local graph clustering algorithms. Solvers were run with multiple time budgets.

## 5.3 Results: maximum clique

Table 1 reports the test set approximation ratio, i.e., the ratio of each solution's cost over the optimal cost. For simple datasets, such as IMDB, most neural networks achieve similar performance and do not violate the problem constraints. On the other hand, the benefit of the probabilistic penalty method becomes clear on the more-challenging Twitter dataset, where training with smooth relaxation losses yields significantly worse results and constraint violation in at least 78% of the instances (see Appendix). Erdős' GNN always respected constraints. Our method was also competitive w.r.t. network RUN-CSP and the best solver, consistently giving better results when optimizing for speed ("fast"). The most accurate method overall was Gurobi, which impressively solved all instances perfectly given sufficient time. As observed, Gurobi has been heavily engineered to provide significant speed up w.r.t. CBC. Nevertheless, we should stress that both solvers scale poorly with the number of nodes and are not viable candidates for graphs with more than a few thousand nodes.

Table 2 tests the best methods on hard instances. We only provide the results for Toenshoff-Greedy, RUN-CSP, and Gurobi, as the other baselines did not yield meaningful results. Erdős' GNN can be seen to be better than RUN-CSP in the training and test set and worse for larger, out of distribution, instances. However, both neural approaches fall behind the greedy algorithm and Gurobi, especially when optimizing for quality. The performance gap is pronounced for small instances but drops significantly for larger graphs, due to Gurobi's high computational complexity. It is also interesting to observe that the neural approaches do better on the training set than on the test set. Since both neural methods are completely unsupervised, the training set performance can be taken at face value (the methods never saw any labels). Nevertheless, the results also show that both methods partially overfit the training distribution. The main weakness of Erdős' GNN is that its performance degrades when testing it in larger problem instances. Nevertheless, it is encouraging to observe that even on graphs of at most 1500 nodes, both our "fast" method and RUN-CSP surpass Gurobi when given the same time-budget. We hypothesize that this phenomenon will be more pronounced with larger graphs.

|  | Training set | Test set | Large Instances |
|---|---|---|---|
| Erdős' GNN (fast) | 0.899 ± 0.064 (0.27 s/g) | 0.788 ± 0.065 (0.23 s/g) | 0.708 ± 0.027 (1.58 s/g) |
| Erdős' GNN (accurate) | 0.915 ± 0.060 (0.53 s/g) | 0.799 ± 0.067 (0.46 s/g) | 0.735 ± 0.021 (6.68 s/g) |
| RUN-CSP (fast) | 0.833 ± 0.079 (0.27 s/g) | 0.738 ± 0.067 (0.23 s/g) | 0.771 ± 0.032 (1.84 s/g) |
| RUN-CSP (accurate) | 0.892 ± 0.064 (0.51 s/g) | 0.789 ± 0.053 (0.47 s/g) | 0.804 ± 0.024 (5.46 s/g) |
| Toenshoff-Greedy | **0.924 ± 0.060 (0.02 s/g)** | 0.816 ± 0.064 (0.02 s/g) | **0.829 ± 0.027 (0.35 s/g)** |
| Gurobi 9.0 (0.1s) | 0.889 ± 0.121 (0.18 s/g) | **0.795 ± 0.118 (0.16 s/g)** | 0.697 ± 0.033 (1.17 s/g) |
| Gurobi 9.0 (0.5s) | **0.962 ± 0.076 (0.34 s/g)** | **0.855 ± 0.083 (0.31 s/g)** | 0.697 ± 0.033 (1.54 s/g) |
| Gurobi 9.0 (1.0s) | **0.980 ± 0.054 (0.45 s/g)** | **0.872 ± 0.070 (0.40 s/g)** | 0.705 ± 0.039 (2.05 s/g) |
| Gurobi 9.0 (5.0s) | **0.998 ± 0.010 (0.76 s/g)** | **0.884 ± 0.062 (0.68 s/g)** | 0.790 ± 0.285 (6.01 s/g) |
| Gurobi 9.0 (20.0s) | **0.999 ± 0.003 (1.04 s/g)** | **0.885 ± 0.063 (0.96 s/g)** | 0.807 ± 0.134 (21.24 s/g) |

Table 2: Hard maximum clique instances (RB). We report the approximation ratio (bigger is better) in the training and test set, whereas the rightmost column focuses on a different distribution consisting of graphs of different sizes. Execution time is measured in sec. per graph (s/g). Pareto-optimal solutions are in bold.

|  | SF-295 | FACEBOOK | TWITTER |
|---|---|---|---|
| Erdős' GNN | **0.124 ± 0.001 (0.22 s/g)** | **0.156 ± 0.026 (289.28 s/g)** | **0.292 ± 0.009 (6.17 s/g)** |
| L1 GNN | 0.188 ± 0.045 (0.02 s/g) | 0.571 ± 0.191 (13.83 s/g) | **0.318 ± 0.077 (0.53 s/g)** |
| L2 GNN | **0.149 ± 0.038 (0.02 s/g)** | 0.305 ± 0.082 (13.83 s/g) | 0.388 ± 0.074 (0.53 s/g) |
| Pagerank-Nibble | 0.375 ± 0.001 (1.48 s/g) | N/A | 0.603 ± 0.005 (20.62 s/g) |
| CRD | 0.364 ± 0.001 (0.03 s/g) | 0.301 ± 0.097 (596.46 s/g) | 0.502 ± 0.020 (20.35 s/g) |
| MQI | 0.659 ± 0.000 (0.03 s/g) | 0.935 ± 0.024 (408.52 s/g) | 0.887 ± 0.007 (0.71 s/g) |
| Simple-Local | 0.650 ± 0.024 (0.05 s/g) | 0.955 ± 0.019 (404.67 s/g) | 0.895 ± 0.008 (0.84 s/g) |
| Gurobi (10s) | **0.105 ± 0.000 (0.16 s/g)** | 0.961 ± 0.010 (1787.79 s/g) | 0.535 ± 0.006 (52.98 s/g) |

Table 3: Cluster conductance on the test set (smaller is better) and execution time measured in sec. per graph. Pareto-optimal solutions are in bold.

## 5.4 Results: local graph partitioning

The results of all methods and datasets are presented in Table 3. To compare fairly with previous works, we evaluate partitioning quality based on the measure of local conductance, $\phi(S) = \text{cut}(S)/\text{vol}(S)$, even though our method only indirectly optimizes conductance. Nevertheless, Erdős' GNN outperforms all previous algorithms by a considerable margin. We would like to stress that this result is not due to poor usage of previous methods: we rely on a well-known implementation [24] and select the parameters of all non-neural baselines by grid-search on a held-out validation set. We also do not report performance when a method (Pagerank-Nibble) returns the full graph as a solution [77].

It is also interesting to observe that, whereas all neural approaches perform well, GNN trained with a probabilistic loss attains better conductance across all datasets. We remind the reader that all three GNNs feature identical architectures and that the L1 and L2 loss functions are smooth relaxations that are heavily utilized in partitioning problems [14]. Furthermore, due to its high computational complexity and the extra overhead that is incurred when constructing the problem instances for large graphs, Gurobi performed poorly in all but the smallest graphs. We argue that the superior solution quality of Erdős' GNN serves as evidence for the benefit of our unsupervised framework.

## 6 Conclusion

We have presented a mathematically principled framework for solving constrained combinatorial problems on graphs that utilizes a probabilistic argument to guarantee the quality of its solutions. As future work, we would like to explore different avenues in which the sequential decoding could be accelerated. We aim to expand the ability of our framework to incorporate different types of constraints. Though we can currently support constraints where node order is not necessarily important (e.g., clique, cover, independent set), we would like to determine whether it is possible to handle more complex constraints, e.g., relating to trees or paths [19]. Overall, we believe that this work presents an important step towards solving CO problems in an unsupervised way and opens up the possibility of further utilizing techniques from combinatorics and the theory of algorithms in the field of deep learning.

# 7   Broader impact

This subfield of deep learning that our work belongs to is still in its nascent stages, compared to others like computer vision or translation. Therefore, we believe that it poses no immediate ethical or societal challenges. However, advances in combinatorial optimization through deep learning can have significant long term consequences. Combinatorial optimization tasks are important in manufacturing and transportation. The ability to automate these tasks will likely lead to significant improvements in productivity and efficiency in those sectors which will affect many aspects of everyday life. On the other hand, these tasks would be otherwise performed by humans which means that such progress may eventually lead to worker displacement in several industries. Combinatorial optimization may also lead to innovations in medicine and chemistry, which will be beneficial to society in most cases.

Our work follows the paradigm of unsupervised learning which means that it enjoys some advantages over its supervised counterparts. The lack of labeled instances implies a lack of label bias. Consequently, we believe that unsupervised learning has the potential to avoid many of the issues (fairness, neutrality) that one is faced with when dealing with labeled datasets. That does not eliminate all sources of bias in the learning pipeline, but it is nonetheless a step towards the right direction.

Finally, we acknowledge that combinatorial optimization has also been widely applied in military operations. However, even though this is not the intention of many researchers, we believe that it is just a natural consequence of the generality and universality of the problems in this field. Therefore, as with many technological innovations, we expect that the positives will outweigh the negatives as long as the research community maintains a critical outlook on the subject. Currently, the state of the field does not warrant any serious concerns and thus we remain cautiously optimistic about its impact in the world.

**Acknowledgements.**   The authors would like to thank the Swiss National Science Foundation for supporting this work in the context of the project "Deep Learning for Graph-Structured Data" (grant number PZ00P2 179981).

## Footnotes

[1]Code available at: https://github.com/Stalence/erdos_neu

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
