[Supplementary Material · CO_supplementary_material.pdf]

## A  Visual demonstration

Figure 1 provides a visual demonstration of the input and output of Erdős' GNN in a simple instance of the maximum clique problem.

a) Input  b) GNN output  c) Integral solution

Figure 1: Illustration of our approach in a toy instance of the maximum clique problem from the IMDB dataset. a) A random node is selected to act as a 'seed'. b) Erdős' GNN outputs a probability distribution over the nodes (color intensity represents the probability magnitude) by exploring the graph in the vicinity of the seed. c) A set is sequentially decoded by starting from the node whose probability is the largest and iterating with the method of conditional expectation. The identified solution is guaranteed to obey the problem constraints, i.e., to be a clique.

We would like to make two observations. The first has to do with the role of the starting seed in the probability assignment produced by the network. In the maximum clique problem we did not require the starting seed to be included in the solutions. This allowed the network to flexibly detect maximum cliques within its receptive field without being overly constrained by the random seed selection. This is illustrated in the example provided in the figure, where the seed is located inside a smaller clique and yet the network is able to produce probabilities that focus on the largest clique. On the other hand, in the local graph partitioning problem we forced the seed to always lie in the identified solution—this was done to ensure a fair comparison with previous methods. Our second observation has to do with the sequential decoding process. It is encouraging to notice that, even though the central hub node has a considerably lower probability than the rest of the nodes in the maximum clique, the method of conditional expectation was able to reliably decode the full maximum clique.

## B  Experimental details

### B.1  Datasets

The following table presents key statistics of the datasets that were used in this study:

|  | IMDB | COLLAB | TWITTER | RB (Train) | RB (Test) | RB (Large Inst.) | SF-295 | FACEBOOK |
|---|---|---|---|---|---|---|---|---|
| nodes | 19.77 | 74.49 | 131.76 | 216.673 | 217.44 | 1013.25 | 26.06 | 7252.71 |
| edges | 96.53 | 2457.78 | 1709.33 | 22852 | 22828 | 509988.2 | 28.08 | 276411.19 |
| reduction time | 0.0003 | 0.006 | 0.024 | 0.018 | 0.018 | 0.252 | – | – |
| number of test graphs | 200 | 1000 | 196 | 2000 | 500 | 40 | 8055 | 14 |

Table 4: Average number of nodes and edges for the considered datasets. Reduction time corresponds to the average number of seconds needed to reduce a maximum clique instance to a maximum independent instance. Number of test graphs refers to the number of graphs that the methods were evaluated on, in a given dataset.

To speed up computation and training, for the Facebook dataset, we kept graphs consisting of at most 15000 nodes (i.e., 70 out of the total 100 available graphs of the dataset).

The RB test set can be downloaded from the following link: `https://www.dropbox.com/s/9bdq1y69dw1q77q/cliques_test_set_solved.p?dl=0`. The latter was generated using the procedure described by Xu [75]. We used a python implementation by Toenshoff et al. [65] that is available on the RUN-CSP repository: `https://github.com/RUNCSP/RUN-CSP/blob/master/`

generate_xu_instances.py. Since the parameters of the original training set were not available, we selected a set of initial parameters such that the generated dataset resembles the original training set. As seen in Table 4, the properties of the generated test set are close to those of the training set. Specifically, the training set contained graphs whose size varied between 50 and 500 nodes and featured cliques of size 5 to 25. The test set was made out of graphs whose size was between 50 and 475 nodes and contained cliques of size 10 to 25. These minor differences provide a possible explanation for the drop in test performance of all methods (larger cliques tend to be harder to find).

All other datasets are publicly available.

## B.2 Neural network architecture

In both problems, Erdős' GNN and our own neural baselines were given as node features a one-hot encoding of a random node from the input graph. For the local graph partitioning setting, our networks consisted of 6 GIN layers followed by a multi-head GAT layer. This depth is consistent across all datasets. We employed skip connections and batch-normalization at every layer. For the maximum clique problem, we also incorporated graph size normalization for each convolution that we found to improve optimization stability. The networks in this setting did not use a GAT layer, as we found that multi-head GAT had a significant impact on the speed/memory of the network without any significant benefits in accuracy to match that cost. Furthermore, locality was enforced after each layer by masking the receptive field. That is, after 1 layer of convolution only 1-hop neighbors were allowed to have nonzero values, after 2 layers only 2-hop neighbors could have nonzero values, etc. The output of the final GNN layer was passed through a two layer perceptron giving as output one value per node. The aforementioned numbers were re-scaled to lie in $[0, 1]$ (using a graph-wide min-max normalization) and were interpreted as probabilities $p_1, \ldots, p_n$. In the case of local graph partitioning, the forward-pass was concluded by the appropriate re-scaling of the probabilities (as described in Section 3.1.3).

## B.3 Local graph partitioning setup

Following the convention of local graph clustering algorithms, for each graph in the test set we randomly selected $d$ nodes of the input graph to act as cluster *seeds*, where $d = 10, 30$, and $100$ for SF-295, TWITTER, and FACEBOOK, respectively. Each method was run once for each seed resulting in $d$ sets per graph. We obtained one number per seed by averaging the conductances of the graphs. Table 3 reports the mean and standard deviation of these numbers.[1]

The volume-constrained graph partitioning formulation can be used to minimize conductance as follows: Perform grid search over the range of feasible volumes and create a small interval around each target volume. Then, solve a volume-constrained partitioning problem for each interval, and return the set of smallest conductance identified.

We used a fast and randomized variant of the above procedure with all neural approaches and Gurobi (see Section C.2 for more details). Specifically, for each seed node we generated a random volume interval within the receptive field of the network, and solved the corresponding constrained partitioning problem. Our construction ensured that the returned sets always contained the seed node and had a controlled volume. For L1 and L2 GNN, we obtained the set by sampling from the output distribution. We drew 10 samples and kept the best. We found that in contrast to flat thresholding (like in the maximum clique), sampling yielded better results in this case.

For the parameter search of local graph clustering methods, we found the best performing parameters on a validation set via grid search when that was appropriate. For CRD, we searched for all the integer values in the [1,20] interval for all 3 of the main parameters of the algorithm. For Simple Local, we searched in the [0,1] interval for the locality parameter. Finally, for Pagerank-Nibble we set a lower bound on the volume that is 10 % of the total graph volume. It should be noted, that while local graph clustering methods achieved inferior conductance results, they do not require explicit specification of a receptive field which renders them more flexible.

### B.4 Hardware and software

All methods were run on an Intel Xeon Silver 4114 CPU, with 192GB of available RAM. The neural networks were executed on a single RTX TITAN 25GB graphics card. The code is executed on version 1.1.0 of PyTorch, and version 1.2.0 of PyTorch Geometric.

### B.5 Pre-trained Models

Pre-trained models of Erdős' GNN for the maximum clique and the constrained minimum cut respectively can be downloaded from the following links:

`https://www.dropbox.com/sh/mdsjrcg9gch8dti/AADWOUUcQMUkChz8SZNXYGnVa?dl=0`

`https://www.dropbox.com/sh/z00ictftyxx3ipf/AADirtiMIwI3_sxCep5GzJf_a?dl=0`

## C    Additional results

### C.1    Maximum clique problem

The following experiments provide evidence that both the learning and decoding phases of our framework are important in obtaining valid cliques of large size.

#### C.1.1    Constraint violation

Table 5 reports the percentage of instances in which the clique constraint was violated in our experiments. Neural baselines optimized according to penalized continuous relaxations struggle to detect cliques in the COLLAB and TWITTER datasets, whereas Erdős' GNN always respected the constraint.

|  | IMDB | COLLAB | TWITTER | RB (all datasets) |
|---|---|---|---|---|
| Erdős' GNN (fast) | **0%** | **0%** | **0%** | **0%** |
| Erdős' GNN (accurate) | **0%** | **0%** | **0%** | **0%** |
| Bomze GNN | **0%** | 11.8% | 78.1% | – |
| MS GNN | 1% | 15.1% | 84.7% | – |

Table 5: Percentage of test instances where the clique constraint was violated.

Thus, decoding solutions by the method of conditional expectation is crucial to ensure that the clique constraint is always satisfied.

#### C.1.2    Importance of learning

We also tested the efficacy of the learned probability distributions produced by our GNN on the Twitter dataset. We sampled multiple random seeds and produced the corresponding probability assignments by feeding the inputs to the GNN. These were then decoded with the method of conditional expectation and the best solution was kept. To measure the contribution of the GNN, we compared to random uniform probability assignments on the nodes. In that case, instead of multiple random seeds, we had the same number of multiple random uniform probability assignments. Again, these were decoded with the method of conditional expectation and the best solution was kept. The results of the experiment can be found in Table 6.

|  | Erdős' GNN | $U \sim [0,1]$ |
|---|---|---|
| 1 sample | $0.821 \pm 0.222$ | $0.513 \pm 0.266$ |
| 3 samples | $0.875 \pm 0.170$ | $0.694 \pm 0.210$ |
| 5 samples | $0.905 \pm 0.139$ | $0.760 \pm 0.172$ |

Table 6: Approximation ratios with sequential decoding using the method of conditional expectation on the twitter dataset. The second column represents decoding with the probabilities produced by the GNN. The third column shows the results achieved by decoding random uniform probability assignments on the nodes.

As observed, the cliques identified by the trained GNN were significantly larger than those obtained when decoding a clique from a random probability assignment.

## C.2 Local graph partitioning

We also attempted to find sets of small conductance using Gurobi. To ensure a fair comparison, we mimicked the setting of Erdős' GNN and re-run the solver with three different time-budgets, making sure that the largest budget exceeded our method's running time by approximately one order of magnitude. We used the following integer-programming formulation of the constrained graph partitioning problem:

$$\min_{x_1,\ldots,x_n \in \{0,1\}} \sum_{(v_i,v_j) \in E} (x_i - x_j)^2 \tag{6}$$

$$\text{subject to} \quad \left(1 - \frac{1}{4}\right) \text{vol} \leq \sum_{v_i \in V} x_i d_i \leq \left(1 + \frac{1}{4}\right) \text{vol} \quad \text{and} \quad x_s = 1.$$

Above, vol is a target volume and $s$ is the index of the seed node (see explanation in Section B.3). Each binary variable $x_i$ is used to indicate membership in the solution set. In order to encourage local solutions on a global solver like Gurobi, the generated target volumes were set to lie in an interval that is attainable within a fixed receptive field (identically to the neural baselines). Additionally, the seed node $v_s$ was also required to be included in the solution. The above choices are consistent with the neural baselines and the local graph partitioning setting.

The results are shown in Table 7. Due to its high computational complexity, Gurobi performed poorly in all but the smallest instances. In the FACEBOOK dataset, which contains graphs of 7k nodes on average, Erdős' GNN was impressively able to find sets of more than $6\times$ smaller conductance, while also being $6\times$ faster.

|  | SF-295 | FACEBOOK | TWITTER |
|---|---|---|---|
| Gurobi (0.1s) | $0.107 \pm 0.000$ (0.16 s/g) | $0.972 \pm 0.000$ (799.508 s/g) | $0.617 \pm 0.012$ (3.88 s/g) |
| Gurobi (1s) | $0.106 \pm 0.000$ (0.16 s/g) | $0.972 \pm 0.000$ (893.907 s/g) | $0.544 \pm 0.007$ (12.41 s/g) |
| Gurobi (10s) | $\mathbf{0.105 \pm 0.000}$ **(0.16 s/g)** | $0.961 \pm 0.010$ (1787.79 s/g) | $0.535 \pm 0.006$ (52.98 s/g) |
| Erdős' GNN | $0.124 \pm 0.001$ (0.22 s/g) | $\mathbf{0.156 \pm 0.026}$ **(289.28 s/g)** | $\mathbf{0.292 \pm 0.009}$ **(6.17 s/g)** |

Table 7: Average conductance of sets identified by Gurobi and Erdős' GNN (these results are supplementary to those of Table 3).

It should be noted that the time budget allowed for Gurobi only pertains to the *optimization time* spent (for every seed). There are additional costs in constructing the problem instances and their constraints for each graph. These costs become particularly pronounced in larger graphs, where setting up the problem instance takes more time than the allocated optimization budget. We report the total time cost in seconds per graph (s/g).

# D    Deferred technical arguments

## D.1 Proof of Theorem 1

In the constrained case, the focus is on the probability $P(\{f(S;G) < \epsilon\} \cap \{S \in \Omega\})$. Define the following probabilistic penalty function:

$$f_p(S;G) = f(S;G) + \mathbf{1}_{S \notin \Omega}\,\beta, \tag{7}$$

where $\beta$ is any number larger than $\max_S\{f(S;G)\}$. The key observation is that, if $\ell(\mathcal{D}, G) = \epsilon < \beta$, then there must exist a valid solution of cost $\epsilon$. It is a consequence of $f(S;G) > 0$ and $\beta$ being an upper bound of $f$ that

$$P(f_p(S;G) < \epsilon) = P(f(S;G) < \epsilon \cap S \in \Omega). \tag{8}$$

Similar to the unconstrained case, for a non-negative $f$, Markov's inequality can be utilized to bound this probability:

$$
\begin{aligned}
P(\{f(S;G) < \epsilon\} \cap \{S \in \Omega\}) &= P(f_p(S;G) < \epsilon) \\
&> 1 - \frac{1}{\epsilon}\mathbb{E}\left[f_p(S;G)\right] \\
&= 1 - \frac{1}{\epsilon}\left(\mathbb{E}\left[f(S;G)\right] + \mathbb{E}\left[\mathbf{1}_{S \notin \Omega}\,\beta\right]\right) \\
&= 1 - \frac{1}{\epsilon}\left(\mathbb{E}\left[f(S;G)\right] + P(S \notin \Omega)\,\beta\right).
\end{aligned}
\tag{9}
$$

The theorem claim follows from the final inequality.

## D.2   Iterative scheme for non-linear re-scaling

Denote by $\mathcal{D}^0$ the distribution of sets predicted by the neural network and let $p_1^0, \ldots, p_n^0$ be the probabilities that parameterize it. We aim to re-scale these probabilities such that the constraint is satisfied in expectation:

$$
\sum_{v_i \in V} a_i p_i = \frac{b_l + b_h}{2}, \quad \text{where} \quad p_i = \mathrm{clamp}\left(c\,p_i^0, 0, 1\right) \quad \text{and} \quad c \in \mathbb{R}.
$$

This can be achieved by iteratively applying the following recursion:

$$
p_i^{\tau+1} \leftarrow \mathrm{clamp}(c^\tau p_i^\tau, 0, 1), \quad \text{with} \quad c^\tau = \frac{b - \sum_{v_i \in Q^\tau} a_i}{\sum_{v_i \in V \setminus Q^\tau} a_i p_i^\tau} \quad \text{and} \quad Q^\tau = \{v_i \in V : p_i^\tau = 1\},
$$

where $b = \frac{b_l + b_h}{2}$.

The fact that convergence occurs can be easily deduced. Specifically, consider any iteration $\tau$ and let $Q^\tau$ be as above. If $p_i^{\tau+1} < 1$ for all $v_i \in V \setminus Q^\tau$, then the iteration has converged. Otherwise, we will have $Q^\tau \subset Q^{\tau+1}$. From the latter, it follows that in every $\tau$ (but the last), set $Q^\tau$ must expand until either $\mathrm{clamp}(c^\tau p_i^\tau, 0, 1) = b$ or $Q^\tau = V$. The latter scenario will occur if $\sum_{v_i \in V} a_i \leq b$.

## D.3   Proof of Theorem 2

Set $b = (b_l + b_h)/2$ and $\delta = (b_h - b_l)/2$. By Hoeffding's inequality, the probability that a sample of $\mathcal{D}$ will lie in the correct interval is:

$$
P\left(\left|\sum_{v_i \in S} a_i - \mathbb{E}\left[\sum_{v_i \in S} a_i\right]\right| \leq \delta\right) = P\left(\left|\sum_{v_i \in S} a_i - b\right| \leq \delta\right) \geq 1 - 2\exp\left(-\frac{2\delta^2}{\sum_i a_i^2}\right).
$$

We can combine this guarantee with the unconstrained guarantee by taking a union bound over the two events:

$$
\begin{aligned}
P\left(f(S;G) < \ell(\mathcal{D}, G) \text{ AND } \sum_{v_i \in S} a_i \in [b_l, b_h]\right) \\
= 1 - P\left(f(S;G) \geq \ell(\mathcal{D}, G) \text{ OR } \sum_{v_i \in S} a_i \notin [b_l, b_h]\right) \\
\geq 1 - P\left(f(S;G) \geq \ell(\mathcal{D}, G)\right) - P\left(\sum_{v_i \in S} a_i \notin [b_l, b_h]\right) \\
\geq t - 2\exp\left(-\frac{2\delta^2}{\sum_i a_i^2}\right)
\end{aligned}
$$

The previous is positive whenever $t > 2\exp\left(-2\delta^2/(\sum_i a_i^2)\right)$.

### D.3.1 Proof of Corollary 1

To ensure that the loss function is non-negative, we will work with the translated objective function $f(S; G) = \gamma - w(S)$, where the term $\gamma$ is any upper bound of $w(S)$ for all $S$.

Theorem 1 guarantees that if

$$\mathbb{E}\left[f(S; G)\right] + P(S \notin \Omega)\beta \leq \ell_{\text{clique}}(\mathcal{D}, G) \leq \epsilon \tag{10}$$

and as long as $\max_S f(S; G) = \gamma - \min_S w(S) \leq \gamma \leq \beta$, then with probability at least $t$, set $S^* \sim \mathcal{D}$ satisfies $\gamma - \epsilon/(1-t) < w(S^*)$.

Denote by $x_i$ a Bernoulli random variable with probability $p_i$. It is not difficult to see that

$$\mathbb{E}\left[w(S)\right] = \mathbb{E}\left[\sum_{(v_i,v_j)\in E} w_{ij}x_ix_j\right] = \sum_{(v_i,v_j)\in E} w_{ij}p_ip_j \tag{11}$$

We proceed to bound $P(S \notin \Omega_{\text{clique}})$. Without loss of generality, suppose that the edge weights have been normalized to lie in $[0, 1]$. We define $\bar{w}(S)$ to be the volume of $S$ on the complement graph:

$$\bar{w}(S) \triangleq \sum_{v_i,v_j\in S} \{(v_i,v_j) \notin E\}$$

By definition, we have that $P\left(S \notin \Omega_{\text{clique}}\right) = P\left(\bar{w}(S) \geq 1\right)$. Markov's inequality then yields

$$P\left(S \notin \Omega_{\text{clique}}\right) \leq \mathbb{E}\left[\bar{w}(S)\right] = \mathbb{E}\left[\frac{|S|(|S|-1)}{2}\right] - \mathbb{E}\left[w(S)\right]$$

$$= \frac{1}{2}\mathbb{E}\left[\left(\sum_{v_i\in V} x_i\right)^2 - \sum_{v_i\in V} x_i\right] - \mathbb{E}\left[w(S)\right]$$

$$= \frac{1}{2}\sum_{v_i\neq v_j}\mathbb{E}\left[x_ix_j\right] + \frac{1}{2}\sum_{v_i\in V}\mathbb{E}\left[x_i^2\right] - \sum_{v_i\in V}\mathbb{E}\left[x_i\right] - \frac{1}{2}\mathbb{E}\left[w(S)\right]$$

$$= \frac{1}{2}\sum_{v_i\neq v_j} p_ip_j + \frac{1}{2}\sum_{v_i\in V} p_i - \frac{1}{2}\sum_{v_i\in V} p_i - \mathbb{E}\left[w(S)\right] = \frac{1}{2}\sum_{v_i\neq v_j} p_ip_j - \mathbb{E}\left[w(S)\right]. \tag{12}$$

It follows from the above derivations that

$$\gamma - \mathbb{E}\left[w(S)\right] + P(S \notin \Omega)\beta \leq \gamma - \mathbb{E}\left[w(S)\right] + \frac{\beta}{2}\sum_{v_i\neq v_j} p_ip_j - \beta\mathbb{E}\left[w(S)\right]$$

$$= \gamma - (1+\beta)\mathbb{E}\left[w(S)\right] + \frac{\beta}{2}\sum_{v_i\neq v_j} p_ip_j$$

$$= \gamma - (1+\beta)\sum_{(v_i,v_j)\in E} w_{ij}p_ip_j + \frac{\beta}{2}\sum_{v_i\neq v_j} p_ip_j. \tag{13}$$

The final expression is exactly the probabilistic loss function for the maximum clique problem.

### D.4 Proof of Corollary 2

Denote by $S$ the set of nodes belonging to the cut, defined as $S = \{v_i \in V, \text{ such that } x_i = 1\}$. Our first step is to re-scale the probabilities such that, in expectation, the following is satisfied:

$$\mathbb{E}\left[\text{vol}(S)\right] = \frac{v_l + v_h}{2}.$$

This can be achieved by noting that the expected volume is

$$\mathbb{E}\left[\text{vol}(S)\right] = \mathbb{E}\left[\sum_{v_i\in V} d_ix_i\right] = \sum_{v_i\in V} d_ip_i$$

495 and then using the procedure described in Section D.2.

496 With the probabilities $p_1, \ldots, p_n$ re-scaled, we proceed to derive the probabilistic loss function
497 corresponding to the min cut.

498 The cut of a set $S \sim \mathcal{D}$ can be expressed as

$$\text{cut}(S) = \sum_{v_i \in S} \sum_{v_j \notin S} w_{ij} = \sum_{(v_i, v_j) \in E} w_{ij} z_{ij}, \tag{14}$$

499 where $z_{ij}$ is a Bernoulli random variable with probability $p_i$ which is equal to one if exactly one of
500 the nodes $v_i, v_j$ lies within set $S$. Formally,

$$z_{ij} = |x_i - x_j| = \begin{cases} 1 & \text{with probability } p_i - 2p_i p_j + p_j \\ 0 & \text{with probability } 2p_i p_j - (p_i + p_j) + 1 \end{cases} \tag{15}$$

501 It follows that the expected cut is given by

$$\begin{aligned} \mathbb{E}\left[\text{cut}(S)\right] &= \sum_{(v_i, v_j) \in E} w_{ij}\, \mathbb{E}\left[z_{ij}\right] \\ &= \sum_{(v_i, v_j) \in E} w_{ij}(p_i - 2p_i p_j + p_j) \\ &= \sum_{(v_i, v_j) \in E} w_{ij}(p_i + p_j) - 2 \sum_{(v_i, v_j) \in E} p_i p_j w_{ij} = \sum_{v_i \in V} d_i p_i - 2 \sum_{(v_i, v_j) \in E} p_i p_j w_{ij}. \end{aligned}$$

We define, accordingly, the min-cut probabilistic loss as

$$\ell_{\text{cut}}(\mathcal{D}; G) = \sum_{v_i \in V} d_i p_i - 2 \sum_{(v_i, v_j) \in E} p_i p_j w_{ij}$$

Then, for any $t \in (0, 1]$, Markov's inequality yields:

$$P\left(\text{cut}(S) < \frac{\ell_{\text{cut}}(\mathcal{D}; G)}{1 - t}\right) > t$$

502 The proof then concludes by invoking Theorem 2

## Footnotes

[1]Please note that the caption of Table 3 incorrectly reports that $d = 25$ seeds were used for the TWITTER dataset and that the best conductance was kept for each graph. The correct procedure is the one described here.