[Reviews · NeurIPS 2020]

Review 1

Summary and Contributions: This paper considers the problem of learning to do combinatorial optimization on graphs. In particular, it focuses on a set of constrained minimization problems: given an objective function and a constraint, find a set of nodes minimizing the objective function subject to the constraint. This is a broad family that includes many NP-hard problems. The goal is to train a neural network such that, when given a new instance of one of these problems, it can efficiently compute a solution satisfying the constraints whose cost is "close" to the optimal cost. This work proposes a novel framework for unsupervised combinatorial optimization on graphs, which is inspired by Erdos's probabilistic proof technique and makes it more likely that the outputs produce a valid solution when compared to previous approaches. Instead of using a differentiable proxy objective as a replacement for the discrete cost, the authors propose a probabilistic method: minimize an upper bound on the expected value (or similar statistic) of the discrete cost. They present two ways to construct this upper bound in the presence of different types of constraints, and prove that the bound ensures that valid low-cost labels can be sampled with high probability. They further describe how, given such a model, it is possible to deterministically extract a labeling with low cost by greedily making choices that do not increase the expected cost. The authors evaluate their approach on two NP-hard tasks: maximum clique and constrained minimum cut. For maximum clique, they show that their method outperforms baseline proxy objective methods, although it does not always beat all learned baselines and generally does worse than special-purpose heuristic algorithms. For constrained minimum cut, their method finds better solutions than heuristics and proxy objective baselines. In all cases, their method generates outputs that satisfy the desired constraints, in contrast to other learned approaches.

Strengths: This approach seems like an exciting and novel approach to solving combinatorial optimization problems. Instead of using a smoothed proxy objective that is not guaranteed to correspond to the original problem, the objective proposed here is directly tied to the quality of the solutions that can be extracted. This ensures that, as long as the model has low enough loss, the resulting outputs obey the constraints and are likely to be low-cost solutions. Another notable property of this approach is that, even though the objective is based on a probabilistic distribution on labelings, it is possible to deterministically extract a solution that obeys the constraints instead of having to repeatedly draw samples. The framework described in section 3 is general enough to apply to a wide variety of problems, and includes theoretical results that describe the conditions under which good solutions are guaranteed. It is unclear exactly how difficult it is in practice to design loss functions that satisfy the requirements of the framework. Even so, this seems like something that could inspire additional research and provide a starting point for solving additional combinatorial optimization problems, or designing other types of upper bound based on a similar probabilistic method. The experiments are thorough, and compare to a wide set of baselines that cover both state-of-the-art learned approaches and heuristic algorithms. Although their model and objective function is not always superior to the other methods, the authors give a good characterization of the strengths and weaknesses of their approach compared to the others.

Weaknesses: Section 3 seems to assume familiarity with a few things that not all readers may know. In particular, in section 3.1.1 the authors use "the probabilistic method" to justify the existence of a label set of a particular small cost. For readers who may be unfamiliar with Erdos's method, this is a bit opaque. It would be nice to have a concrete example here, or a more intuitive explanation of what the method is and how it works. As another case, the "method of conditional expectation" is explained only very briefly, and could benefit from a more intuitive explanation or a worked-out example. There are a few minor errors in the presentation of the method, which I have noted in the "correctness" section below.

Correctness: I believe the statement of Theorem 1 as written is incorrect. Here is a counterexample: let f(S;G)=0, beta=1, and sigma be the empty set. It is obvious that the probability of sampling a valid label set is zero, since there are no valid label sets. But Theorem 1 states that, with probability at least t (for any t), the sampled label set is a valid label set. The problem seems to be that there is a missing condition on the relationship between the loss and beta (and possibly also t). The proof relies on the loss being strictly less than beta, but this is not required in the actual statement of the theorem, which only constraints the function f to be less than beta. This is obscured by the change in notation between the proof, which uses epsilon and constrains it to be less than beta, and the theorem, which uses t and thus doesn't mention anything about epsilon. Another error: in section 3.2, I believe the comparison has been flipped. To generate an example with a small cost, it would be better to make the choice of v_i that leads to the lower expected cost, not the higher expected cost, right? So line 168 should have "<" instead of ">". These issues seem fairly easy to fix, and I don't think either of them will cause any problems for the rest of the paper.

Clarity: The paper is well written and was enjoyable to read. I though the discussion of prior work was clear, and the approach was well motivated. My only concern is that the method may be difficult to understand for people who are unfamiliar with the probabilistic method or the method of conditional expectation; it would be good to provide a bit more explanation of these in the final paper.

Relation to Prior Work: The paper does a good job of describing prior work in this space, and contextualizing their approach relative to this prior work.

Reproducibility: Yes

Additional Feedback: ==== Edit: after author response ===== I am glad my suggestions were helpful, and look forward to seeing the updated paper.


Review 2

Summary and Contributions: [Post-rebuttal] Great job again; a truly excellent paper!! This paper establishes a connection between Erdos' Probabilistic Method and learning heuristics for graph optimization problems. The family of graph optimization targeted here involves constrained binary problems with node-based decision variables; many problems can be cast in this form, including Max Clique and Graph Partitioning, which are explored as case studies here. The key idea is that a parametric mapping, here a Graph Neural Network (GNN), from instances of a graph optimization problem to node probabilities can be trained, without knowledge of optimal solutions, such that high-quality solutions are assigned high probabilities. This is in stark contrast to both ML-based optimization methods that require optimal solutions in the training phase, and reinforcement learning methods that typically have high sample complexity in combinatorial optimization problems. The main step involves constructing an appropriate (unsupervised) loss function that has the desired probabilistic property above. The authors demonstrate how that can be done for the two case studies in question, and similar analytical derivations can be obtained for other graph problems with a reasonable amount of work. Given a trained model, the probabilistic method provides a simple iterative procedure to "decode" a good solution from the predicted node probabilities. Experimentally, the proposed Erdos GNN performs well compared to recent ML-based combinatorial heuristics, classical heuristics and two exact solvers (depending on the case study). Remarkably, Erdos GNN rarely decodes solutions that violate both problems' constraints. Overall, this is an excellent paper that should be accepted.

Strengths: - A novel, principled framework for learning heuristics for graph optimization problems: the paper establishes a deep connection between the probabilistic method and learning probabilistic heuristics for graph optimization. Unlike many recent papers in this space which are rather incremental in combining GNN with reinforcement learning in various ways, this paper proposes a fresh, fundamentally new perspective. - Interest to NeurIPS: This paper is likely to lead to substantial interest from the NeurIPS and optimization communities, as it involves nice ideas from theoretical computer science, deep learning and discrete optimization. - Impressive experimental results, showing that Erdos' GNN rarely produces infeasible solutions and often produces high-quality solutions for both Clique and Graph Partitioning. - Solid experimental setup: Comparison against various (very recent) ML and non-ML based algorithms, including exact solvers.

Weaknesses: - Further experiments: I would've liked to see more experimental results on additional graph problems. This could drive your point further by showing more examples of how the loss function can be constructed. - Scalability bottleneck: GNN training can only scale up to some level. Thoughts on whether your method can be scaled up despite this limitation? - Architecture choices/tuning: In lines 219-227, how did you select this particular architecture? B.2 in the appendix did not clarify this. There are many ingredients to your architecture but it is difficult to figure out what's necessary and what's not. Perhaps an ablation study or detailed results on the validation sets can help here.

Correctness: - Lines 277-278: "Nevertheless, we should stress that both solvers scale poorly with the number of nodes and are not viable candidates for graphs with more than a few thousand nodes.": You may be referring to scalability w.r.t. finding feasible solutions, whereas Gurobi is an exact solver who's also working to certify optimality. There are ways to encourage MIP solvers like Gurobi to focus on finding feasible solutions, e.g. by setting MIPFocus = 1 in Gurobi: https://www.gurobi.com/documentation/9.0/refman/mipfocus.html.

Clarity: The paper is beautifully written and largely flawless.

Relation to Prior Work: Related work is discussed in sufficient depth and compared against if suitable.

Reproducibility: Yes

Additional Feedback: Nice work on the Broader impact section!


Review 3

Summary and Contributions: This paper presents a novel method to solve graph-based combinatorial optimisation with deep neural networks. The proposed approach adopts the probabilistic method and transforms the objective of classical combinatorial optimisation problems to continuous and differentiable surrogate loss. Then a deep neural net applied to optimise the probability density function on the vertices of the graph that minimises the surrogate loss function. Finally, the answers are selected by sampling from the learned probability distribution.

Strengths: - The idea of using a probabilistic method to derive surrogate loss from classical CO problems and optimise it with deep neural nets is novel; - The paper is well-written and easy to read. The proposed method is simple and straightforward, and achieves a good result on the tested tasks; - Although the performance of the presented approach is still not comparable to classic approximation-based CO tools, it does outperform the other DNN-based methods significantly.

Weaknesses: - The technique of converting constraints into loss function is ad-hoc, which needs to be designed specifically for the CO task; - The algorithm will work when the probability distribution $\mathcal{D}$ is easy to find, and the surrogate loss is convex. When the derived loss function is non-convex, then there would still be an exponential search space for the initialisation of the neural net, which could even be larger than the original problem space (i.e., it may perform worse than solving the original problem directly with classic methods). I think the author should state in the paper about the limitation of the proposed technique.

Correctness: The proposed approach is reasonable. I only roughly checked the derivations and proofs in the supplementary, but they appear to be correct.

Clarity: This paper is well written and did a good job of clearly explaining the motivation and the derivations of the proposed method.

Relation to Prior Work: This paper has covered a broad range of references, and clearly discussed the difference to previous approaches.

Reproducibility: Yes

Additional Feedback: - Does the time cost in Table 2 include the training time of $g_\theta$? - Eq 4 and line 150: What is $a_i$? What is its relationship to $v_i$? - Line 116 says $\mathcal{D}$ is defined "over sets", which is ambiguous because it could refer to the power set of $V$ or $V$ itself. ----------------------------- In the rebuttal the authors answered my question about the generality of this approach, and clarified that in practical the neural nets are robust for minimising the loss. I think this is a very interesting work and I recommend for acceptance.


Review 4

Summary and Contributions: This work presents a unsupervised method for combinatorial optimization problems on graphs based on learning probabilistic distributions over sets (nodes) and using these distributions for reasoning about the probability that nodes belong to the solution set on the graph. The main contribution is in the theoretical development of this probabilistic method for identifying the probability of nodes in being members of such a set. Specifically, the probabilistic loss is problem-specific and must be developed for each problem such that the distributions learned by the GNN produce useful (constraint-obeying) distributions over nodes. Two examples of applying the method for finding this loss are included in the paper (min cut and max clique). The method is evaluated on maximum clique identification and local partitioning (min cut) and compared against some neural approaches, algorithmic approaches, and solvers based on integer programing. Compared to neural approaches, the method matches or improves performance and does not exhibit any constraint violations.

Strengths: The work tackles an important field, CO problems on graphs, which are generally computationally hard and specifically the ability to learn in an unsupervised setting avoids the problem of generating labels for these hard problems. Furthermore, the presented method ensures that the constraints are met, avoiding issues introduced by methods that make use of relaxations to facilitate learning. The main contribution is in the general framework for loss functions of such CO problems.

Weaknesses: One major advantage of neural approaches is the promise of scalability/speed up. However, experiments on unseen large instances shows weak performance (both in speed and accuracy) where the Toenshoff-Greedy method outperforms all methods on both metrics. For graph partitioning, the smooth-relaxation GNNs (L1, L2 loss) methods perform comparably to Erdos GNN but an order of magnitude faster on these particular datasets considered.

Correctness: Claims appear to be well supported by the proofs provided in the appendices and the empirical analysis is clear. One especially important aspect of the empirical study is the inclusion of highly optimized solvers that serve as very strong baselines.

Clarity: The paper is well written and the development of the ideas are laid out nicely and well founded in the motivation

Relation to Prior Work: Structured neural approaches to combinatorial optimization problems are well cited and the relationship between them and the proposed work is discussed clearly.

Reproducibility: Yes

Additional Feedback: Thank you to the authors for their rebuttal that provides answers to the questions I raised in my original review about greedy methods and RUN-CSP.

[Author Response · NeurIPS 2020]

First, we would like to thank the reviewers for their well thought out and detailed feedback on our work.

**Reviewer 1:** (a) We acknowledge the importance of improving the accessibility of our work to the machine learning community and thus we will include an explanatory figure (in Section 3) along with background on the Erdős probabilistic method and the method of conditional expectation. (b) In terms of correctness, both the statement of theorem 1 and the direction of the inequality at line 168 will be updated accordingly for the final version of the paper. Again, we thank the reviewer for pointing out those mistakes.

**Reviewer 2:** (a) We decided to focus our attention on two problems that have different types of constraints to illustrate the flexibility of our approach. As discussed in Sec. 6, certain constraints would be more complicated to handle, e.g., imposing a tree or path structure on the solutions. We agree that solving more problems is essential to demonstrate the generality of our framework and it is one of our current priorities with extending this work. (b) Regarding scalability, indeed GNNs can be computationally expensive in practice. However, this is an active field of research and recent works indicate that it is possible to scale GNN training to millions of nodes [Bojchevski et al., 2020, Rossi et al., 2020]. Another appealing possibility is that of "emergent generalization" as it has been described by Joshi et al. [2019]. That is, a GNN trained on smaller instances of a problem could be trained to perform well on larger instances at test time. We are optimistic that such engineering and conceptual developments in the field of GNNs will yield practically efficient methods over time. (c) Regarding our GNN architecture and tuning, our decisions follow standard conventions in the field: batch and graph size normalization as well as skip connections have been shown to improve performance. Furthermore, our use of GIN and GAT was based on their prominence in the GNN literature as experimentally successful modules. These observations and more interesting practical details about GNNs can be found in the work by Dwivedi et al. [2020]. (d) Regarding the lines 277-278 where we discuss solver scaling, we have indeed used the default Gurobi settings (MIPFocus=0) and therefore our statement will be adjusted to reflect that.

**Reviewer 3:** (a) The reviewer's summary is largely accurate, however, we would like to emphasize that our solutions are obtained through the method of conditional expectation instead of sampling. (b) We recognize that the loss needs to be derived for each particular problem. At the same time, the probabilistic penalty formulation allows for a rather general and principled way of constructing loss functions for CO problems. (c) We would also remark that the loss functions that we have derived are not generally convex/concave, and that neural-networks/SGD have been successful at providing solutions to problems with non-convex objectives. Ultimately, our view is that the ability of our method to yield good solutions will largely depend on how well the neural network will minimize the loss. This, in turn, will depend on how well the neural networks' inductive biases match the problem constraints. Finally, in practice we have observed that, if the architecture can minimize the loss efficiently, then it can also be carefully tuned to do so consistently regardless of the weight initialization (i.e., with different random seeds).

**Reviewer 4:** (a) It is indeed the case that for larger clique instances the greedy method is more efficient. Greedy methods strike a nice balance between accuracy and speed. Yet, on the real world datasets it is clear that the greedy methods are inferior when it comes to accuracy. The main reason why the greedy algorithm outperforms the neural methods on the hard instance dataset (RB) is the time budget limitation that we have imposed in order to have comparable time costs for all the methods. Given a sufficiently large time budget, both our method and RUN-CSP can match the accuracy of the greedy heuristic (some experiments along those lines an be found in the paper by Toenshoff et al. [2019]). Additionally, the use of a suitable greedy heuristic necessitates expert knowledge, which is something a neural approach would circumvent by fitting the data distribution. (b) For partitioning, our method is slower than the smooth relaxations that we compare against, but consistently achieves superior accuracy. In addition, as it can be seen in Table 5 of the supplementary material, smooth relaxations generally struggle to yield solutions that obey more complicated constraints, as it is the case in the maximum clique problem. The advantage of our approach is that it can consistently satisfy constraints while also achieving good performance. Currently, our sequential decoding module induces the largest computational overhead, although the number of loss function evaluations is still linear on the number of nodes in the worst case. Improving the runtime of our decoding module is a subject we are actively investigating.

# References

A. Bojchevski, J. Klicpera, B. Perozzi, A. Kapoor, M. Blais, B. Rózemberczki, M. Lukasik, and S. Günnemann. Scaling graph neural networks with approximate pagerank. *arXiv preprint arXiv:2007.01570*, 2020.

V. P. Dwivedi, C. K. Joshi, T. Laurent, Y. Bengio, and X. Bresson. Benchmarking graph neural networks. *arXiv preprint arXiv:2003.00982*, 2020.

C. K. Joshi, T. Laurent, and X. Bresson. On learning paradigms for the travelling salesman problem, 2019.

E. Rossi, F. Frasca, B. Chamberlain, D. Eynard, M. Bronstein, and F. Monti. Sign: Scalable inception graph neural networks. *arXiv preprint arXiv:2004.11198*, 2020.

J. Toenshoff, M. Ritzert, H. Wolf, and M. Grohe. Run-csp: Unsupervised learning of message passing networks for binary constraint satisfaction problems. *arXiv preprint arXiv:1909.08387*, 2019.


[Meta-Review · NeurIPS 2020]

This is a surprising, novel, and principled framework for unsupervised ML-based combinatorial optimization. It translates the Erdos' Probabilistic Method for combinatorial optimization into a learning framework, producing impressive results. The paper should be improved following suggestions discussed in the rebuttal, but this should be straightforward. Overall, I'll quote R2: "Unlike many recent papers in this space which are rather incremental in combining GNN with reinforcement learning in various ways, this paper proposes a fresh, fundamentally new perspective."